# Methodological Innovations for Establishing Cemetery Spatial Databases—A UAV-Based Workflow Helping Small Communities

**Márton Pál [1,\*] and Edina Hajdú [1,2]**

[1] Institute of Cartography and Geoinformatics, ELTE Eötvös Loránd University, H-1117 Budapest, Hungary

[2] Doctoral School of Earth Sciences, ELTE Eötvös Loránd University, H-1117 Budapest, Hungary; hajdu.edina@inf.elte.hu

[\*] Correspondence: pal.marton@inf.elte.hu

**Abstract:** Various modern large-scale mapping techniques have already been introduced in earth sciences, cadastral mapping, and the agricultural sector. These methodologies often use remotely sensed data to compile various analogue or digital cartographic products as well as spatial databases. However, the mapping of cemeteries and standards for establishing a spatial database for them have rarely been published, and there is no definite method for this purpose in Hungary yet. We have compiled a methodology based on mapping experiences in three sample areas in the Roman Catholic Archdiocese of Eger in Hungary that are church properties. The initial UAV-based fieldwork orthomosaics were processed with a CV (computer vision)-based script that vectorised grave contours. After fieldwork, which included the recording of the deceased people's names and their dates of birth and death in the case of all graves, a spatial database was created pairing each polygon with the corresponding personal data. A map was also generated from the results of the survey. The cartographic product and the database fulfil legal requirements and give hints for cemeteries regarding further planning. The developed method is capable of making mapping and database building easier—not just in the case of graves, but with other rectangular objects, too.

**Keywords:** cemetery databases; CV; UAV; photogrammetry; large-scale mapping

## 1. Introduction

Cemeteries are usually places and symbols of religion, culture, and personal emotions—and their role in public life is unquestionably essential [1]. Sometimes, these properties are considered urban green spots that are recognized as "mirrors of society" because they show attributes of the structure of society, customs related to burials and architecture, and religious composition [2,3].

Because of these factors, various science sectors have examined cemeteries, as they are good sample areas of natural and cultural processes, traditions, and phenomena. Studies in geosciences (especially geology) [4], archaeology [5], anthropology [6], medicine and pathology [7,8], and some projects related to cartography and GIS [9–11] have also included these manmade objects.

The examination of cemeteries regarding GIS and cartography purposes is a relatively new field of study. Although there have been some initial projects on applying GIS processes and workflows in modern cemetery research works, where the cultural, historic, or traditional attributions of cemeteries were examined with digital mapping techniques [11–13], most of the studies focus on establishing the fundamental outlines of digital cemetery GIS systems [14,15]. According to the scientific literature, no complete mapping methodologies have been presented yet for smaller communities that help them fit governmental regulations or just manage land use with cadastral tools. However, the current legal modifications in Hungary set various standards for estate owners (mainly

municipalities or various religious denominations) that force them to make a detailed, large-scale map and a complex spatial database of every cemetery, which should be capable of aiding further planning and editing the cadastral system of each graveyard and the corresponding infrastructure.

The goal of this study is to present a methodology that can be used for (1) making large-scale cemetery maps and (2) providing a detailed spatial database in the background—with the aid of UAV-based remote sensing and computer vision (CV). The results of an initial drone survey were processed with an open-source drone mapping software (OpenDroneMap). The generated orthomosaics were analysed by a program written in Python (using the OpenCV and GDAL/OGR libraries) to vectorise grave contours from the rasters. We found some false detections and small missing graves (mostly false positives resulting from detection size threshold values or other objects like pavement stones or large cement flowerpots). After filtering these vectorisation mistakes manually (we cleared the unnecessary objects or added the missing ones in QGIS), fieldwork took place to collect personal information from each grave of the sample cemeteries. Having the digitized set of objects, each record from the field was assigned to the corresponding grave. This stage of the spatial database was completed by using the DEM (which was compiled from the UAV survey photos with a resolution of 5 cm) to acquire contour lines and digitise other field objects based on satellite imagery or the generated orthomosaics.

This manuscript presents the workflow in detail and provides an opportunity to apply this kind of survey and database methodology in any modern-day cemeteries in the world. The main result of this study is the automatic vectorisation of graves that makes map editing less time-consuming and gives further development strategies to make maps of other object types, too. Workflows and algorithms of quasi-similar purposes (e.g., counting graves) mainly use supervised or unsupervised classification for grouping and categorizing objects [16,17], but there is no exact literature for vectorizing graves automatically. We have chosen a CV-based approach due to the short algorithm running time and the relatively didactic programming scheme. Although detailed cemetery cadastral surveys and even online maps are available in big cities in Hungary (especially in Budapest), these solutions are unnecessary for villages and settlements with small cemeteries without any historical relevance. With this workflow, the establishment of lightweight but legally appropriate digital cemetery databases are made much easier for small communities that cannot invest in expensive digitization projects.

*Sample Areas: The Cemeteries of Hevesaranyos, Istenmezeje, and Váraszó in the Archdiocese of Eger*

The Archdiocese of Eger (Latin: Archidioecesis Agriensis) is a Roman Catholic archdiocese in Northern Hungary (Figure 1a). Its centre is Eger, where the archbishop sits. The cathedral basilica is also here. The population of the archdiocese is around 1,260,000 and the ratio of Roman Catholics is 54.4% (~685,000 people). It was established by the first Christian king of Hungary (Saint Stephen I) in 1004, so it was one of the first dioceses in the country. It is patronised by St John the Evangelist. The Diocese of Eger was promoted to the Metropolitan Archdiocese of Eger in 1804 when Ferenc Fuchs became the first archbishop [18,19]. Its extent varied through the centuries; its current form was established by Pope Saint John Paul II in 1993, when the religious administration borders of Hungary were rationalised. The current archbishop has been Csaba Ternyák since 2007.

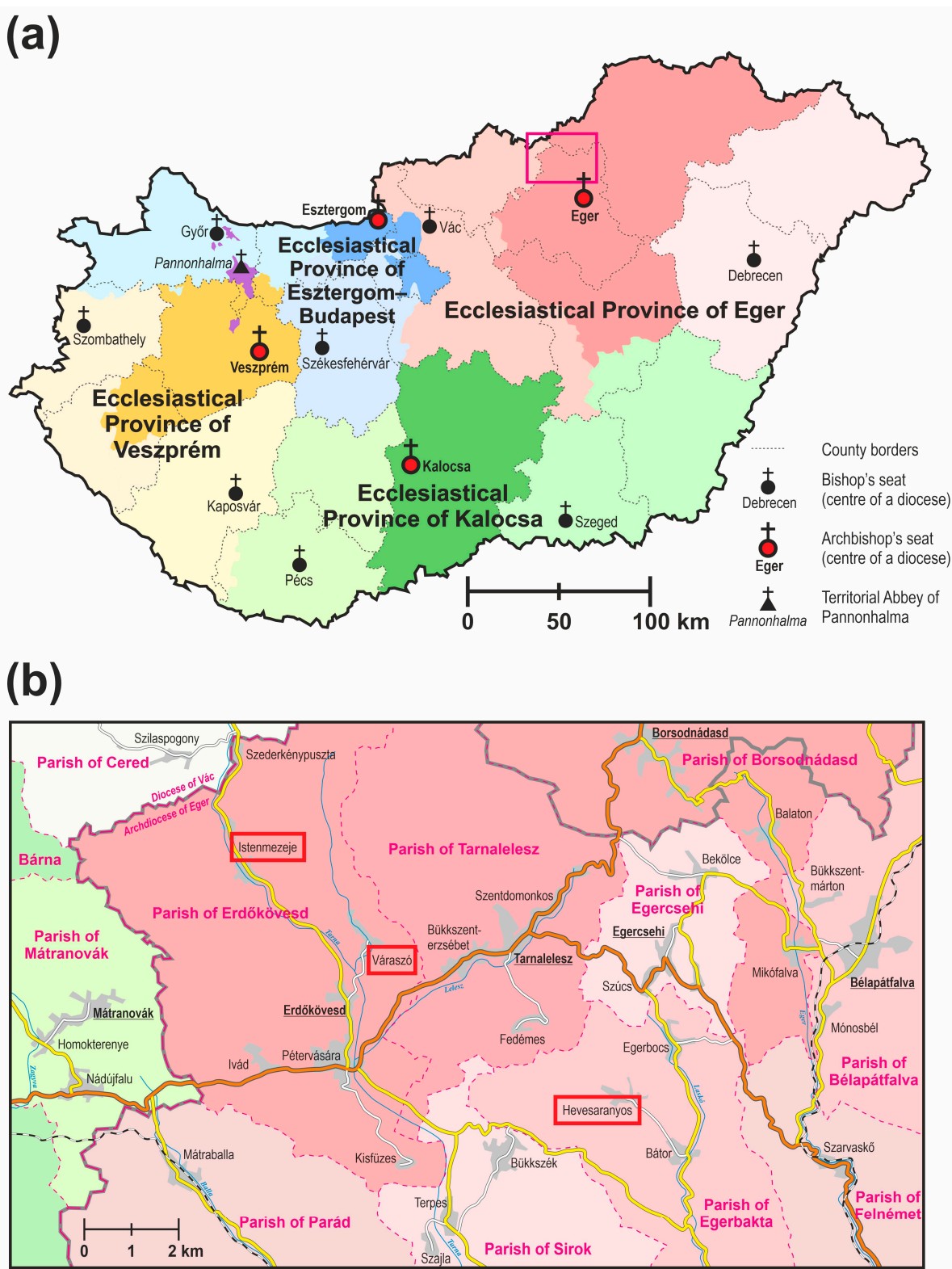

**Figure 1.** (**a**) The ecclesiastical administrational units of the Roman Catholic Church in Hungary (the sample areas are in the magenta rectangle)—the various colour shades mean the territorial extent of dioceses within an ecclesiastical province. (**b**) The Roman Catholic administration of the surroundings of the sample settlements (in red rectangles)—red shades mean the different parishes in the Archdiocese of Eger, while green shades mean the different parishes in the Diocese of Vác; parish settlements are underlined.

The area of Hungary is split into ecclesiastical provinces led by archdioceses. Roman Catholic dioceses and archdioceses are divided into smaller ecclesiastical entities, called deaneries (or decanates), that are chaired by a dean. These deaneries are also divided into smaller territorial units, called parishes. In the past, most settlements (and each church in settlements with multiple churches) formed distinct parishes, but currently, a small number of villages (and distinct churches) form a single parish (mainly due to the lack of ecclesiastic personnel). The main church of the parish is called the parish church, where the parish priest sits. Every other church community in the parish area is called a filial church: simply meaning that these are dependent on the parish church and the parish priest [18].

The estate ownership of cemeteries in Hungary is not uniform: they can be owned either by the municipality or the corresponding religious community (or church). Graveyards in the Catholic rural areas of Hungary are mostly owned by the Roman Catholic Church, and the management of the infrastructure is one of the duties of the local deanery.

Our sample cemeteries are in the Archdeanery of the Cathedral in two distinct deaneries and parishes (Figure 1b). Istenmezeje and Váraszó belong to the Parish of Erdőkövesd in the Deanery of Parád, while Hevesaranyos belongs to the Parish of Bátor (that is served by the Parish of Egerbakta) in the Deanery of Eger. All three cemeteries are owned by the Church, and all of them lack a detailed map and database, which are essential due to legal regulations.

The cemeteries are located in hilly areas in the foothills of the Mátra and Bükk Mountains, so there are mainly significant relief changes. The sample villages are relatively small, with a population of less than 1500 each. Due to the small number of inhabitants, past social circumstances, and relief differences, rural cemeteries were not thoroughly planned: the graves were mostly established ad hoc until the near past. One of the reasons for the stricter regulation is this and the poor infrastructure (e.g., the lack of tap water and asphalt roads inside the cemeteries). The three sample areas show different levels of structuredness (Figure 2). The graves in Istenmezeje are placed on the side of a steep hill, not following any planned system. Hevesaranyos cemetery is located on a hilltop allowing moderately systematic grave placements. The cemetery of Váraszó is also on a hilltop—but it has the most planned grave layout (although this is the oldest village of these). These sample areas with different attributions were chosen to establish and test our mapping methodology.

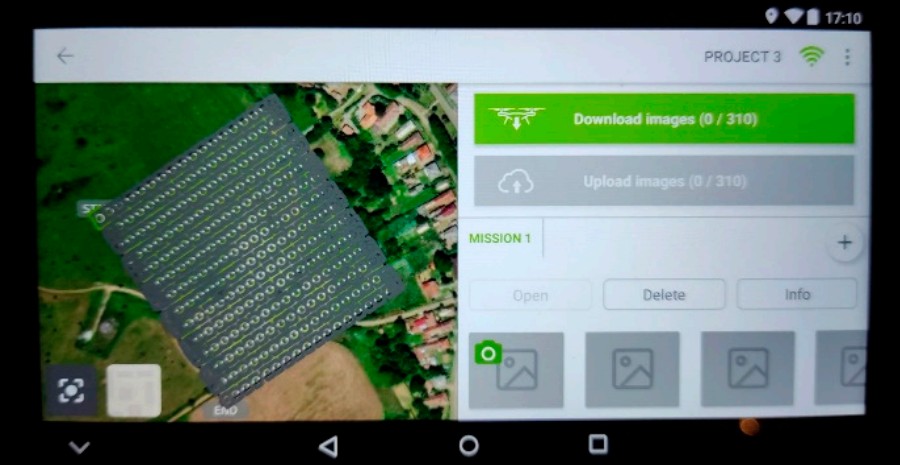

**Figure 2.** The executed mission plan with the position of the taken photos in Pix4D Capture. The photo was taken from the screen of the Phantom 4 Pro+.

## 2. Methodology

The workflow consisted of four distinct steps:

1.  UAV survey of the cemeteries—for generating orthomosaics;
2.  Fieldwork—photo-based documentation of the graves for database records;
3.  Automatic vectorisation—a program for faster polygon drawing;

4.    Map and database compilation—a GIS workflow for producing the results.

*2.1. UAV Survey*

The first step before taking the survey was to fit the regulations concerning drone operations issued by the EU and the Hungarian government. The legal situation was not simple, because Váraszó and Istenmezeje fall into the Tarnavidék Nature Protection Area—because of this, distinct permission was needed. As we worked with drones heavier than 120 g and equipped with visual sensors, we needed strict flying certificates, as these cemeteries are adjacent to inhabited areas. These licenses allow the pilot to fly near or over inhabited areas. After receiving all permissions and airspace booking (which is also required by the Hungarian legislation [20]), the flying missions were planned considering weather conditions. As we needed to book airspace 30 days before the flight for 7 days, it was impossible to go to the field when the weather conditions were simply good—we had a 7-day time interval when we could go. If the weather was bad (it rained, or the wind speed was high) we had to book another time interval for 30 days later. The "best" weather for a planned mission flight is when there is no wind, the temperature is above 0 °C, and it is overcast (in this case, there are no shadows).

The circumstances were different for all three missions. In Istenmezeje, the fieldwork was carried out in March, with cold temperatures and a cloudy sky (disturbed by a flurry). The Hevesaranyos mission took place in May with warm and partly cloudy weather (the changing cloudiness conditions forced us to carry out the mission again after the first attempt—that time, there were no clouds during flying). We worked in July in Váraszó with a clear sky and extremely hot temperatures. Two UAVs were used: a DJI Matrice 2010 V2 RTK (manufacturer: DJI, Nanshan, Shenzhen, China) with a Zenmuse X5S RGB sensor (in Istenmezeje) and a DJI Phantom 4 Pro+ (in Hevesaranyos and Váraszó).

The DJI Matrice 210 V2 RTK is a quadcopter designed mainly for industrial use. Its unfolded size is 883 × 886 × 427 mm, and it weighs 4.91 kg. The drone flies with two Li-Po batteries which allow for 25–30 min of flight time depending on the payloads. Our model has two gimbals, but we only used one of them for the X5S RGB sensor [21].

The DJI Phantom 4 Pro+ is a much smaller UAV with a size of 196 × 290 × 290 mm and a weight of 1.388 kg. It can be used for many purposes from hobby photo taking and filming to entry-level industrial purposes. With a single battery, it can fly for 30 min. It has one gimbal with one built-in sensor that cannot be changed [22].

The cause for changing the UAV after the first mission was that the Matrice 210 was more difficult to transport and manage in the field. Additionally, the charging time was longer than in the case of the Phantom 4. However, there is no significant difference between their sensors, as both were only used to take photos. The Zenmuse X5S has a maximum resolution of 20.8 MP without optical zoom [23]. The Phantom 4′s sensor also has 20M effective pixels with just digital zoom [22]. So, the use of the smaller drone was a choice based on practical criteria.

All missions were carried out at a height of 35 m. The cause for this relatively high value is the relief differences in the case of all three cemeteries. As the sensors almost have the same physical attributes, the resolution and other visual characteristics of the resulting orthomosaics were nearly the same. This is why only the light conditions resulting from the differences in cloudiness and clear sky can cause any differences.

We used the DJI Pilot software during the Istenmezeje flight (as this is the default software of the remote controller), while in the other two villages, we planned and executed the missions using the freeware Pix4D Capture software, which showed higher reliability (Figure 2, Phantom 4 Pro+ is not compatible with the DJI Pilot app). The Phantom 4′s default software is DJI Go, but before this project, we experienced some bugs within the software that led to dangerous UAV movements. Therefore, we decided to use the open-source alternative of Pix4D based on professional recommendations and drone compatibility. With the Matrice, we did not use ground control points (GCPs) as it was equipped with an RTK

module. When we carried out our missions with the Phantom 4 drone, we used 5 GCPs in the cemeteries that we measured with an external RTK GPS device.

The surveys were successful—but the experiences with the Phantom 4 device and Pix4D Capture combo were more promising. Side and frontal overlaps of 80% were used during all three flights. The Istenmezeje mission resulted in 402 images, the Váraszó cemetery had 310, and the Hevesaranyos area had 189 RGB pictures. We generated a 2D orthomosaic from the captured images with the open-source OpenDroneMap software (in the WebODM application [v2.0.3], Figure 3) with a resolution of 5 cm. This way, all three cemeteries had their aerial mosaics. Additionally, a Digital Elevation Model (DEM) was also derived for the further extraction of contour lines with the same 5 cm accuracy (using QGIS 3.14).

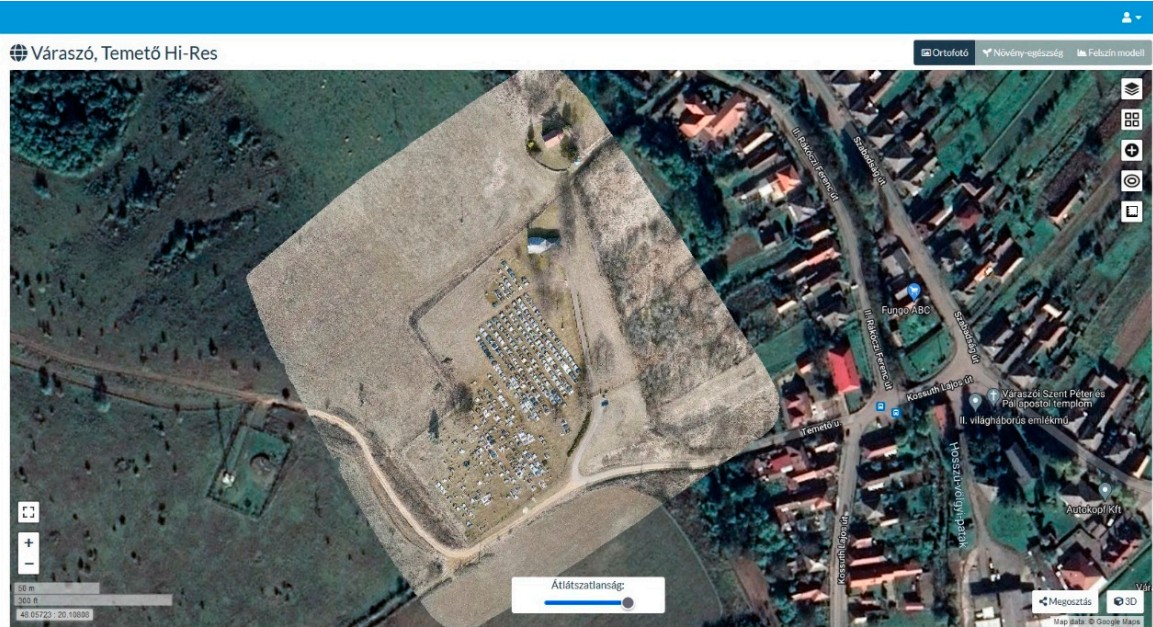

**Figure 3.** The OpenDroneMap's WebODM platform was used to produce orthomosaics (the figure was modified after [24]).

*2.2. Automatic Vectorisation*

We created a workflow in Python for the delineation of graves. We used the PyCharm IDE, in which the necessary dependencies could be installed (these can be found in the Python script in the appendix of this paper). The script relied on two main pillars: the grave edge recognition with OpenCV and vectorization, and the feature extraction of GDAL/OGR.

The first step was to modify the colours of the image and filter it for easier CV interpretation. This included the following conducted processes:

- Converting the default OpenCV BGR colour scheme to RGB;
- Converting RGB to grayscale;
- Applying noise filtering on the grayscale image.

This was followed by Canny edge detection. The purpose of this algorithm is to check if the pixels with the same direction have larger or smaller intensity values than the pixels that are just under process. The result of this was an image including the most important edges of the orthomosaic: edges of pavements, flower beds, roads, buildings, steps, and other natural or manmade objects. We pre-defined a segment length (1 m) that was used to draw out only longer ones than this value. (A maximum can also be easily added, but it was unnecessary for the areas we worked with.) The bounding rectangles can be easily produced from continuous edges.

Of course, the automatic process works with some errors. These happen if some false-positive edges are detected (that are not grave edges but come from other objects like pavements and buildings), if there are graves with discontinuous edges (e.g., graves in bad condition), or other geometry issues. The bounding rectangle of these objects is not accurate and does not represent a grave. These errors were handled in a later step (after vectorization) manually.

OpenCV has a function to follow the subprocess results and draw images after each algorithm. This way, each step can be supervised and the algorithm values can be modified based on the subprocess results.

Depending on the spatial characteristics of the graves (size, distribution, the recognized edges), we should test the parameters of creating bounding rectangles. If the threshold is too high, many graves will be omitted; if the threshold is too low, many insignificant features are also bounded. The image containing the rectangles can be saved with GDAL as a GeoTIFF file. With the use of GDAL/OGR, the exact pixels of rectangles can be extracted, and the area closed by them can be turned into a vector polygon. Later, all these polygons can be exported into a GIS vector format.

The ratio of accuracy regarding detected gravestone objects was 70–80% in all three cemeteries. Because of this, manual editing was also needed after image processing. This automatisation workflow (based on [24]) can be summarised in the following steps and used functions:

- Modifying colour models to fit OpenCV (cv2.COLOR_RGB2BGR; cv2.COLOR_BG R2GRAY);
- Noise filtering and edge detection (cv2.blur, cv2.Canny);
- Searching for contours (cv2.findContours);
- Determining bounding rectangles of contours (cv2.approxPolyDP);
- Monitoring the process (cv2.imshow);
- Setting the file format, projection, raster band, and writing as an array (gdal.GetDriver ByName, Create, SetGeoTransform, SetProjection, GetRasterBand, WriteArray),
- Converting to vector (ogr.GetDriverByName, CreateDataSource, outDataSource.Create Layer, gdal.Polygonize). The Python script is appended to this.00. *paper.

### 2.3. Fieldwork

We carried out the required data collection as part of our extensive fieldwork on-site. Each person's name in the cemetery was registered using photos taken on the spot. This work usually lasted for approximately half a workday.

Every grave was identified and paired with the data on the orthomosaic using the images. The fieldwork was also an opportunity to check the validity of the vectorised dataset. We printed the 50% opaque vectorized rectangles over a satellite image. Mistakes could be identified this way: graves that were falsely drawn could be corrected, while the ones that had been missed were put in the polygon dataset. We digitized around 15–20 graves additionally after the fieldwork. However, in most cases, these did not have a gravestone, just a cross and a heap of earth.

### 2.4. Map and Database Compilation

In the office work phase, we matched all personal data with the corresponding grave polygon using the photos and the orthomosaic. This was carried out using an index number (the photo number assigned to the corresponding grave). To ensure the detailedness of the project, we also included birth and death dates where possible. In some cases, death years were not carved into the stone. This was important due to cadastral purposes.

In some situations, graves were in such bad condition that we could not note down any information from them (especially in the case of very old and unmaintained ones; Figure 4). However, most of the objects were in good condition (Figure 5), but some of these were also illegible.

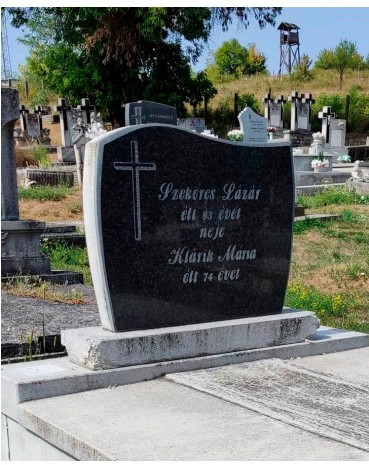

**Figure 4.** A grave in good condition; it is well-readable.

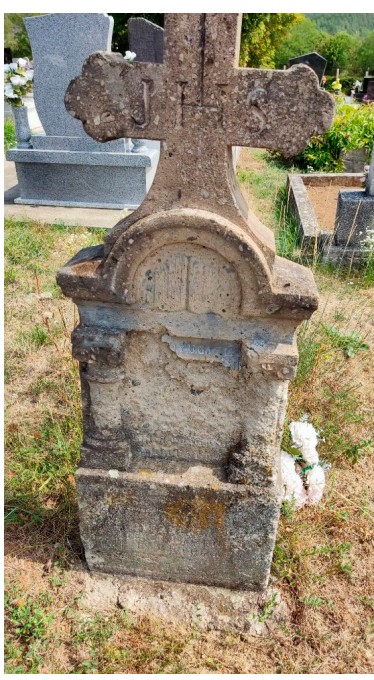

**Figure 5.** An old gravestone in bad condition. The names and numbers are eroded and not readable.

## 3. Results

We provide two distinct results for the parishes: an analogue (paper) map and the digital version (GIS and graphical files) with a spatial database (the attribute scheme can be found in Section 3.2) that can be easily modified in the future.

### 3.1. Cemetery Maps

All field data, a topographic base map, and the vectorized datasets (graves and DEM-based contour lines) were used to compile the cemetery maps.

We also added roads, pavements, trails, fences, buildings, and other (mostly manmade) objects (like benches, taps, and crucifixes—these were extracted from the topographic base and the orthomosaic). The area of the cemeteries was divided into parcels that did not include more than 200 graves. Their delineation was determined by mostly "evident" boundaries: larger, distinct groups of graves; natural borders, like trees or bushes; and roads. The freeware software QGIS 3.28 was used for mapmaking. We compiled two map types: an overview map that helps to locate each distinct parcel (Figure 6) and one for

the whole cemetery (Figure 7). The compiled maps were placed at the entrance of the cemeteries.

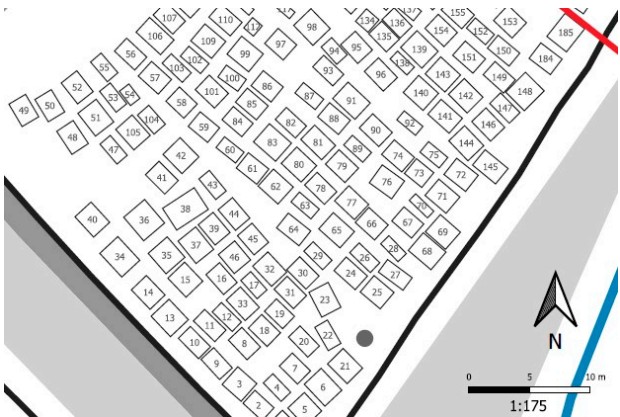

**Figure 6.** An excerpt of the map of the first parcel in the Istenmezeje cemetery (the extent of the excerpt is marked with a green rectangle in Figure 7).

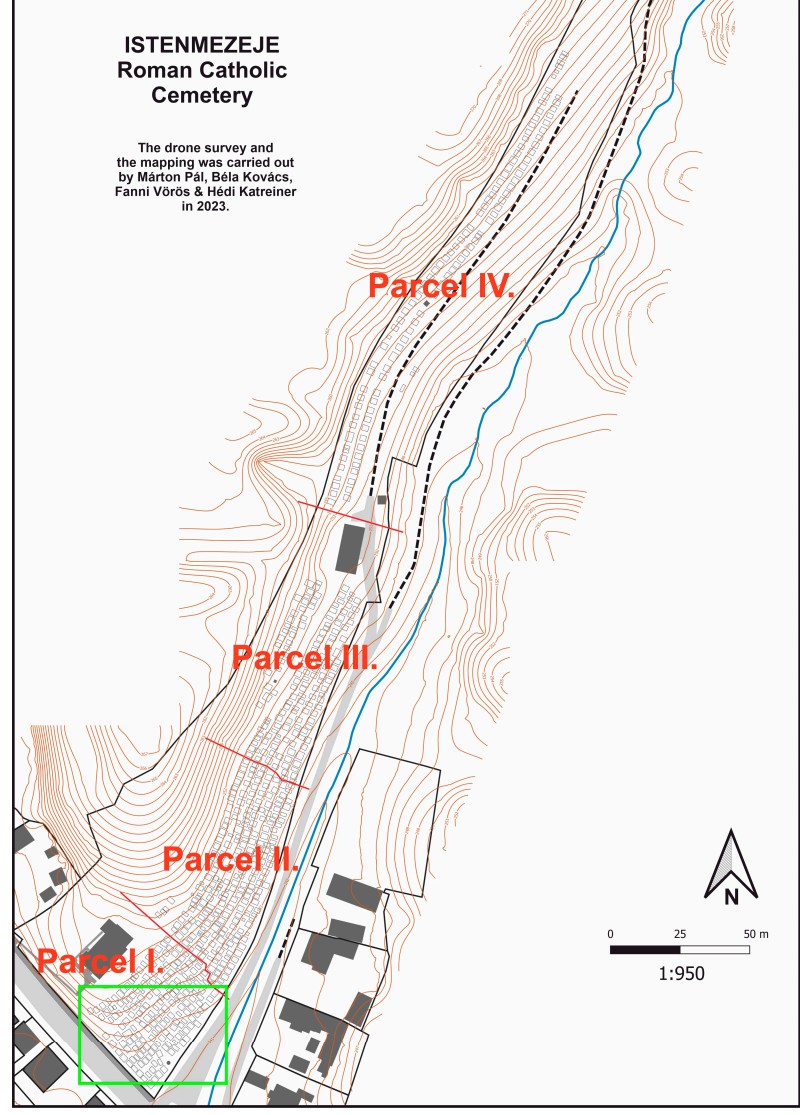

**Figure 7.** The overview map of Istenmezeje cemetery.

*3.2. Cemetery Databases*

The purpose of the database (and the digital version of the map) is to provide an opportunity for smaller communities to continuously monitor cemetery changes on their own. They can easily record new graves and personal information in the Excel-based datasheets, while newly built objects can also be added to the graphic database using simple editing tools. The thorough administration of graves and buried people is required to meet the standards of public health regulations. We inserted the following attributes in the textual database:

- The identification number of the grave;
- The parcel of the grave;
- The names and the birth and death dates of the buried people.

The database in Istenmezeje consists of data on 680 graves and 1421 people, in Váraszó on 582 graves and 1367 people, and in Hevesaranyos, on 559 graves and 1298 people. The worksheets can be continuously filled with newer records in the future.

## 4. Conclusions and Discussions

In this study, we propose a new methodology for small communities to meet cemetery regulations on spatial and textual databases in cemeteries. The most important factors and findings of this project are the following:

**Data acquisition in the field has become much easier with the application of drones**. This UAV- and CV-based methodology offers a budget-friendly alternative for small communities to meet the legal requirements of cemetery cadastral surveys. Furthermore, the GIS and the textual database can be easily complemented later if needed.

**The manual vectorization is replaced by an automatic method that reduces the working hours of database and map compilation**. The drawing of more than 1600 graves just for this exploratory project would have been much longer than automatic vectorization with manual correction. In the case of more cemeteries, this methodology can be usefully utilized to reduce labour needs.

**The consistency of data is ensured overall because data validity and personal data of buried people are controlled manually**. Although the vectorization is automatic, its results are checked manually to filter out mistakes, incorrectly detected objects, or missing graves. In addition, names and dates are collected manually from the graves based on field photos, and textual data are paired manually to the vectorized graves. This is also a cross-check to find random errors.

**The textual database and the graphic form can easily be updated by non-experts, too**. Updating the vector layer can be carried out by adding new grave rectangles to the GPKG file; however, a person is needed who is competent in editing such files. The textual worksheet is easier to manage: the personal data of deceased people are the only data that should be added.

Despite the modern workflow and the multiple opportunities to validate generated and acquired data, there are some important factors that we should take into consideration when mapping, because they may result in false data and unsuccessful database building.

The mapping personnel operating the UAV must follow the legal background of UAV missions that are stricter than the general directive of the European Union [19,25]. As the presented workflow is to meet cemetery cadastral laws, the UAV personnel is also expected to be legal.

The correct orthomosaic production requires expertise in planned drone missions and strong IT hardware for photogrammetric processing. The UAV pilot must know the sample area to a certain level to be able to satisfactorily plan the mission height and light conditions. Additionally, during mission planning, the personnel also must take into consideration the battery capacity, the dimensions of the cemetery, and the overlap ratios to be able to produce good and vectorizable results. Strong hardware is also needed for processing the images; in particular, VRAM (video RAM) and RAM are needed for working with hundreds of images.

Although we used an open-source photogrammetric software (OpenDrone Map—WebODM), its installation may be difficult for those who do not have an IT background (there are alternative solutions, but they are not for free). Nonetheless, the photogrammetric task can be undertaken by using any software that can produce orthomosaics and DEMs from surveyed data.

To carry out the mapping itself using original and derived data, we also need to know some basic GIS and cartography—although these are not complex maps and they do not contain very dense information.

The presented workflow can be developed according to several distinct directions:

The accuracy ratio of the automatization workflow of 70–80% could be enhanced by experimenting with other solutions and UAV survey conditions. The importance of various heights, image resolutions, flying speeds, light conditions, and overlaps can be examined to define the ideal conditions for computer vision to delineate objects.

The automatization methodology is planned to be extended to other objects (not just quasi-rectangular, but also with other shapes)—e.g., applying artificial intelligence. Other object classes with regularity (like roofs, pools, cars, buildings, etc.) could also be mapped following the methodology of this study. Approaches rooted in AI may also enhance the results; e.g., supervised classification connected to CV may highlight new research directions.

The real-time sensing of objects with computer vision can be applied in national defence and safety projects for detecting specific types of objects (e.g., illegal buildings). The military and law enforcement application of this CV-based detection approach can also have a role in area monitoring for specific objects. Tunnel entrances, shelters, or other manmade structures could be delineated and mapped.

**All the mapping workflow (except for the fieldwork) could be carried out using FOSS (free and open-source software) tools**. The whole methodology presented in this study was carried out by using only open-source means of data processing and GIS editing. This is an important factor, as small communities usually do not have the proper funding to invest in expensive software and tools.

**Author Contributions:** Conceptualization, Márton Pál; methodology, Márton Pál and Edina Hajdú; software, Márton Pál and Edina Hajdú; validation, Márton Pál; data curation, Márton Pál and Edina Hajdú; writing—original draft preparation, Márton Pál; writing—review and editing, Edina Hajdú; visualization, Márton Pál; project administration, Márton Pál; funding acquisition, Márton Pál. All authors have read and agreed to the published version of the manuscript.

**Funding:** The authors were supported by project no. TKP2021-NVA-29, which was implemented with the support provided by the Ministry of Innovation and Technology of Hungary from the National Research, Development and Innovation Fund, financed under the TKP2021-NVA funding scheme.

**Data Availability Statement:** All data are available here: https://mercator.elte.hu/~marchello/cemetery/data.zip. Accessed: 12 February 2024.

**Conflicts of Interest:** The authors declare no conflicts of interest. The funders had no role in the design of the study; in the collection, analyses, or interpretation of data; in the writing of the manuscript; or in the decision to publish the results.

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
