# Peer review of "Methodological Innovations for Establishing Cemetery Spatial Databases—A UAV-Based Workflow Helping Small Communities"

_ijgi, doi:10.3390/ijgi13020057_

Round 1
Reviewer 1 Report
Comments and Suggestions for Authors
Thank you for submitting your interesting work. The methodology, results and discussion/conclusion sections are quite weak as only abstract-level details are provided. The manuscript's structure needs to align with journal guidelines and the typical academic paper structure.
1. Introduction
· Line 36-37: The statement, “Although there have been some initial projects about applying GIS processes and workflows in modern (not historic or prehistoric) cemetery research”. What kind of research, please elaborate on it.
· Line 56: The statement, “ After filtering the vectorisation mistakes…..”
· What kind of mistakes were found? Please describe in deatil.
· How the mistakes were corrected?
· Line 59: The statement, “ ……. the spatial database was completed by using the DEM…. What was resolution of the DEM?
2. Methodology
· Line 116: For better understanding of the readers, please describe the main characteristics of the UAV?
· Line 131: How many ground control points were used for UAV survey?
· Line 150: Please attach Python script as appendix
3. Results
Line 210: There is no mention of attribute schema. Please add with complete details
4. Discussion and Conclusion
· Please align the discussion section with the findings
Comments on the Quality of English LanguageModerate editing required
Author Response
Dear Reviewer,
Thank you for your valuable comments and suggestions. We have elaborated the manuscript following your advices. The Python script is also appended to the revised submission that is uploaded in the system.
Yours sincerely,
Márton Pál
Reviewer 2 Report
Comments and Suggestions for Authors
The paper describes a novel methodology developing cemeteries databases based in UAV capture. The work seems promising but the paper has many flaws which need to be worked on if it is to be re-submitted.
Some of the structural flaws include:
1. No serious review of the concepts and technologies used, as well as no similar projects described, listed and analysed. No review section is provided.
2. Methodological description is very incomplete, not acceptable. No description of the options analysed and decisions taken is provided.
3. Results lack structure and evidence.
4. Limitations and details of future work could be improved.
5. Included maps must be shown complete and including legends. Figure 1 a) does not have a legend. Figure 6 is part of a map. The complete map should also be included with the zoomed bit highlighted for context. The map in figure 7 is not in english.
6. Figures' quality must be improved. Figure 2 is a photo of a screen!! Figure 3 is a screen capture of a software which is in not in English. If you cannot at all provide the information in english, more information needs to be provided as legend. The same goes for any figures which include text that is not in english.
7. Link https://mercator.elte.hu/~marchello/cem-287 etery/data.zip has an error and does not work.
If I were to try this methodology, I would not have enough information to check the work done. Also, the tests need to be extended to include additional possibilities not considered in this paper.
The description also needs to be improved and detailed. The work should be extended and re-submitted or submitted in other venues (such as conferences of the topic) as work in progress.
Detailed comments are provided in the attached file.

Comments on the Quality of English LanguageAuthor Response
Dear Reviewer,
Thank you for your valuable comments and suggestions. We have elaborated the article in line with your advices. Let me react to some of your concerns:
- No serious review of the concepts and technologies used, as well as no similar projects described, listed and analysed. No review section is provided.
We have not found pre-existing literature for this kind of process. However, we have added a paragraph comparing our methodology to other methods mainly applied in large cemeteries or in those of historical relevance. - Notes regarding the figures:
We have provided extended captions. For figure 2.: we had to take a photo of the screen during field work, as there is no option within the Pix4D Capture software for making a print screen. he photo was taken of the remote control display of the DJI Phantom 4 Pro+ UAV. - Link availability:
Unfortunately, the link was unavailable due to the line break. We have corrected the hyperlink and now the content can be downloaded.
Other changes and refinements can be traced in the revised manuscript.
Yours sincerely,
Márton Pál
Reviewer 3 Report
Comments and Suggestions for Authors
The objective of research was to elaborate a methodology that can be used for making large-scale cemetery maps and for providing a detailed spatial database in the background with the aid of UAV-based remote sensing and computer vision.
In the introduction should be added arguments for the usage of proposed methodology in practice showing its concrete added value. Usually cemetery administrations provide maps and databases with more detailed information than offered by the authors.
Some cited references seems not be relevant to the research, especially sources number 1, 4, 6, 7, 8.
The methods are not adequately described for practical usage of the proposed methodology. The main step of the methodology is analysis of generated orthomosaics by a program written in Python (mentioned in lines 54-55) using the OpenCV and GDAL/OGR libraries, to vectorize grave contours from the rasters. This program was not shared by the authors for open access or sufficiently described in the manuscript to be implemented by readers.
Comments on the Quality of English LanguageText written in other languages should be translated to English.
Author Response
Dear Reviewer,
Thank your for your valuable comments and suggestions. The manuscript has been elaborated in line with your advices.
Best regards,
Márton
Reviewer 4 Report
Comments and Suggestions for Authors
The aims and methods are sufficiently and clearly documented. The topic, i.e. the presented workflow, is of practical relevance.
Comments on the Quality of English LanguageThe paper is clearly written.
Please check line 242: "Data field data acquisition"
Author Response
Dear Reviewer,
Thank you for your comments. The mentioned line was corrected.
Best regards,
Márton
Round 2
Reviewer 1 Report
Comments and Suggestions for Authors
Thank you for improving quality of the manuscript. It is suggested to separate Discussion and Conclusions section and preferably into paragraphs instead of bullets.
Author Response
Dear Reviewer,
Thank you again for your valuable contribution.
Best regards,
Márton Pál
Reviewer 3 Report
Comments and Suggestions for Authors
In the current version of the manuscript, the authors have made significant corrections, supplemented missing content, and better structured the content.
Author Response
Dear Reviewer,
Thank you again for your valuable contribution.
Best regards,
Márton